# Pleuroparenchymal Fibroelastosis: A Review with a Focus on a Non-Infectious Complications after Hematopoietic Stem Cell Transplant

**DOI:** 10.3390/biomedicines11030924

**Published:** 2023-03-16

**Authors:** Patrick Arndt

**Affiliations:** Division of Pulmonary, Allergy, Critical Care, and Sleep Medicine, Department of Medicine, University of Minnesota, Minneapolis, MN 55455, USA; arndt108@umn.edu

**Keywords:** hematopoieitic stem cell transplant, pleuroparenchymal fibroelastosis, interstitial lung disease, biomarkers

## Abstract

Pleuroparenchymal fibroelastosis (PPFE) is a rare disease that is currently classified as an idiopathic interstitial pneumonia. Although originally described as an idiopathic disease, PPFE has now been identified as a rare complication following hematopoietic stem cell transplant (HSCT). Unlike other pulmonary complications after HSCT, PPFE occurs very late after transplant. Etiologies for PPFE after HSCT remain to be fully established. Infections and adverse effects to alkylating chemotherapy have been suggested as possible causes. In several cases, there is an association of PPFE with bronchiolitis obliterans syndrome after HSCT, suggesting that PPFE may be another manifestation of pulmonary chronic graft versus host disease after HSCT. Algorithms have been designed to assist in confirming a diagnosis of PPFE without the need for a surgical lung biopsy, however at present, no biomarker is established for the diagnosis or to predict the progression of disease. Presently, there is no current therapy for PPFE, but fortunately the disease progresses slowly in most patients.

## 1. Introduction

Subpleural fibroelastosis is a rare interstitial lung disease that has been described for many years but remains poorly understood. The diagnosis initially gained attention through the description by Amitani et al. of an upper lobe predominant pleural-based disease, initially termed pulmonary upper lobe fibrosis (PULF), that later carried his name, Amitani disease [1,2,3,4]. This disease was felt to be distinct from the radiologically similar process, apical cap disease. A similar disease presentation to that described by Amitani was later described by Frankel et al., who subsequently named it pleuroparenchymal fibroelastosis (PPFE) [5]. In these initial descriptions, PPFE was defined as being an upper lobe process with an undefined etiology and was accordingly considered an idiopathic lung disease [1,2,3,4]. Although considered to be a rare disease, the number of cases reported has increased since its initial description, with PPFE now included as an idiopathic interstitial pneumonia (IIP) in the most recent American Thoracic Society classification of IIP’s in 2013 [6]. As stated, however, there was an increase in the reported number of individuals affected with PPFE in the last 10 years, though the published case series remain small, with the largest series of patients numbering just over a 100, and most studies being much smaller [7,8,9]. As such, due to the size of the published case series, advancement in the understanding of PPFE as it relates to diagnosis, etiology, treatment, and outcomes remains limited. 

PPFE was initially described as an idiopathic disease, however it has now been associated with several clinical conditions, including systemic sclerosis, rheumatoid arthritis, the result of frequent respiratory infections, a complication of alkylating chemotherapy or radiotherapy, and as a complication following lung or hematopoietic stem cell transplant (HSCT) [7,10,11,12,13,14]. It can also be seen co-existing with other IIP’s, including chronic hypersensitivity pneumonitis and usual interstitial pneumonia [12,14]. Although PPFE has been associated with other diseases, including after HSCT and lung transplantation, the trigger inducing the onset of PPFE in patients with these conditions has not been identified. This review will discuss what is currently known regarding PPFE as a complication after HSCT, highlighting discussions regarding establishing a diagnosis, potential biomarkers, current treatments, and clinical outcomes and prognosis. This review will also highlight areas that require additional investigation and research.

## 2. PPFE after HSCT

PPFE was initially described as an idiopathic disease, however it was later observed in patients who had undergone a hematopoietic stem cell transplant (HSCT) and is now considered to be a long-term non-infectious pulmonary complication (LTNIPC) after HSCT. This was first described in 2011 in a small series of 4 patients, that all coincidentally had bronchiolitis obliterans [10]. Since that original description, several other reports have described the development of PPFE after HSCT [11,12,13,15,16,17]. Studies that have directly examined PPFE after HSCT are listed in Table 1. 

Overall, it appears that PPFE after HSCT is rare, with an overall prevalence of PPFE after HSCT of 0.28–3.3% [15,16,17]. Interestingly, this number is much lower compared to those who have undergone lung transplantation and subsequently develop PPFE, where the prevalence is 7.54% [15]. The reason for this distinct difference between those who have undergone a HSCT versus those who underwent a lung transplant is currently unknown. Unlike other LTNIPC’s after HSCT that typically occur within 100 days to 1–2 years after HSC, PPFE is a very late pulmonary complication after HSCT, with a median time to diagnosis of 6.9 years, with a wide time range of 0.25 to 17.9 years to the time of diagnosis [10,11,13,15,17]. However, as seen in the listed time range to diagnosis, occasionally, earlier cases are observed [10,11,13,15,17]. The possibility exists that those with an earlier manifestation have a separate distinct disease process or a specific targetable etiologic factor. Most commonly, PPFE is diagnosed in patients who have undergone allogeneic transplant, but there have also been cases observed in those who underwent autologous transplant [10,15]. This observation is important as it may allow the identification of potential etiologies causing PPFE after HSCT, and in particular if PPFE is a manifestation of chronic graft versus host disease, and thereby has a similar pathophysiology as the classic late pulmonary complication of HSCT, being bronchiolitis obliterans syndrome (BOS). This would place PPFE as one form of lung-associated chronic graft versus host disease (cGvHD). In support of this is that various studies have shown that post-HSCT patients who develop PPFE also have established lung-associated cGvHD (i.e., bronchiolitis obliterans syndrome (BOS)) [10,11,13,16,18]. The co-existence of BOS in patients diagnosed with PPFE has been shown to be high (47–100%) [10,11,13,18]. This association has also been true in the post-HSCT PPFE patients followed in our clinic (unpublished observations). This suggests that the underlying etiology and ongoing pathophysiology for BOS and PPFE may be similar. However, as the etiology for BOS, similar to PPFE, remains unknown, this remains only speculative at this time. In addition, as BOS occurs in 5–10% of post-HSCT patients, with PPFE only a much less common post-HSCT complication (a prevalence after HSCT of just 0.2–0.3%), taken together, this suggests that there may be separate entities with distinct etiologies. 

## 3. Etiology of PPFE after HSCT

The underlying etiology of PPFE in post-HSCT patients is unknown, however several potential etiologies have been described. The etiologies that have been suggested to induce PPFE after HSCT are similar to those described above for the non-idiopathic, non-HSCT PPFE cases [7,12]. Two of these have a particular relevance to HSCT, specifically an adverse effect of chemotherapy or radiation therapy used to treat the underlying disease or during the conditioning regimen for HSCT and respiratory infections after transplant, particularly viral infections [14,15,18,20]. Implicated chemotherapeutic agents include the alkylating agents, specifically cyclophosphamide and carmustine [13,14,18,20]. With the overall incidence of PPFE after HSCT being very low, with most patients receiving alkylating hemotherapy during induction for HSCT, the association of PPFE with chemotherapy exposure in patients who have undergone HSCT remains to be fully determined, but a clear cause and effect seems to be less likely. This is based on the number of patients who received alkylating chemotherapy and the low prevalence of PPFE after HSCT, estimated at 0.2–0.3%, as well as the long duration between alkylating chemotherapy exposure and the onset of PPFE, which is several years. Treatment with radiation during the conditioning period has been discussed as a potential cause for later developing PPFE, however as only a small fraction of the patients who are treated with whole-body irradiation during preparation for HSCT go on to develop PPFE, this seems less likely as a sole cause for the later development of PPFE [13,14]. A second relevant proposed etiology is a complication after a respiratory infection. The types of respiratory infections implicated in causing PPFE are vast, and include all infectious agents including bacterial, viral, fungal, and atypical Mycobacteria [14,15,18,21,22]. A note should be made of the recent descriptions of a potential association of atypical Mycobacteria infection and PPFE. In addition to the more common infectious agents, infection with an atypical Mycobacterium has been recently suggested as an etiology for PPFE in non-HSCT patients [21,22]. This has been shown both with the finding of a co-association of PPFE in a series of infected patients with atypical Mycobacteria and in the observation that a few patients with PPFE also had a co-existing infection with atypical Mycobacteria [21,22]. In the two large retrospective studies, the prevalence of PPFE in those with an atypical Mycobacterial infection was found to be 11.9–26.3% [21,22]. In the study by Aonoi et al., only 1 of 845 patients (0.001%) described in their study underwent a HSCT prior to developing PPFE [22]. However, to date, no studies in post-HSCT patients have extensively examined a possible association of an infection with atypical Mycobacteria and PPFE. Taken together, as it relates to infection as a cause of PPFE in post-HSCT patients, no specific infection has been proven to induce PPFE and no animal model exists showing a connection between a pulmonary infection and the later development of PPFE. The cases described implicating recurrent infections in post-HSCT patients as a cause for PPFE lack a detailed description of the type of infection as well as the frequency, duration, and timing of the respiratory infection prior to the development of PPFE [14,15]. Respiratory infections are common after HSCT, particularly early after transplant when patients are significantly immunocompromised. Dissuading respiratory infections as a cause for PPFE after HSCT is two-fold. Firstly, respiratory infections, and in particular viral infections, are very common, whereas PPFE is very rare, and secondly, the timing from a purported respiratory infection and PPFE in the studies is quite broad, sometimes measured in years, with no detailed description of the severity or duration of the respiratory infection [15,18]. In addition, the possibility of an adverse reaction to the medications used in the treatment of pulmonary infections and the development of PPFE has not been investigated. To date, there have been no large prospective, retrospective, or longitudinal follow-up studies examining a potential connection between respiratory infections and PPFE in patients post-HSCT. 

## 4. Clinical Presentation

The clinical presentation of PPFE after HSCT is similar to those cases due to other etiologies or those considered to be idiopathic. Patients commonly present with progressive shortness of breath and cough [4,10,12,16]. Typically, the shortness of breath is slowly progressive, and the cough is dry in character. The time from the onset of shortness of breath to the diagnosis of PPFE has a wide range (0.5 to 6 years), with a mean of 2.5 years from the time of symptom onset to establishment of a diagnosis [4,7,16]. Patients may also have chest pain that is pleuritic in nature, but this is not seen in all patients [10,12,14]. Extrathoracic symptoms are unusual. On physical exam, patients typically have a low body mass index (BMI) with a history of weight loss [4,12,14]. The presence of a low BMI is a diagnostic criterion used for PPFE and one that assists in differentiating PPFE from other forms of IIP [4,12,14]. Due to the upper lobe predominance of PPFE, with the upper lobe fibrosis causing hilar retraction, patients with PPFE can develop a flattened chest, termed platythorax [4,12,14]. The degree of platythorax can be calculated by determining the longitudinal versus transverse ratio at the level of the 6th thoracic vertebra [14]. On lung auscultation, crackles can be heard, but their presence is less common compared to patients with idiopathic pulmonary fibrosis, with crackles only heard in patients with PPFE who have co-existing lower lobe involvement with usual interstitial pneumonitis (UIP) or non-specific interstitial pneumonitis (NSIP) [4]. This is also true for nail clubbing, which is seen in only 4–25% of patients with PPFE [14]. 

As the presenting symptoms of shortness of breath, cough, and chest pain are non-specific, the diagnosis of PPFE is commonly suspected based on the unique findings observed on chest imaging. Chest radiographs show pleural thickening in the upper lobes, with retraction of the hila upwards due to volume loss, with subpleural fibrosis consisting of bands of fibrous tissue extending from the pleura to the hilum [4,7,16,23]. In addition to pleural thickening and subpleural fibrotic changes, pneumothoraxes may also be seen on chest imaging at the time of presentation or during the disease course [8,12,14,18]. Pneumothoraxes are common in these patients, occurring in 25–75% of PPFE patients during their illness [8,14,18]. A special note should be made that patients with PPFE are at a high risk of developing a pneumothorax after surgical lung biopsy, with these being persistent after the procedure [4,14]. In contrast, pleural effusions are rare in patients with PPFE. On chest CT imaging, pleural thickening with bands of connective tissue directed toward the hilum are described. Although initially described as an upper lobe disease, the lower lobes can also be involved in PPFE, but less commonly [4,9,11,23]. When lower lobes are involved, an equal distribution between the upper and lower lobes can be observed in up to one third of the patients [24]. The radiologic pattern in the lower lobes can be similar to that in the upper lobes, with pleural thickening and fibrous bands extending toward the hilum; however, findings in the lower lobes may also show diffuse interstitial thickening, suggestive of a concomitant interstitial lung disease [4,7,8,11,13,16]. Pathologic findings from the lower lobes in affected patients may be consistent with PPFE but also commonly show a UIP or NSIP pattern [4,7,8,9,11,13]. This occurs in 32–75% of all patients with PPFE and is not specific to those who have undergone HSCT. Differentiating the presence of UIP (clinically idiopathic pulmonary fibrosis) co-existing with PPFE can be challenging. The upper lobe findings of fibrotic bands and subpleural thickening on imaging are atypical for IPF, and therefore would suggest a diagnosis of PPFE. The only way, however, to fully exclude IPF in the lower lobes is by an open lung biopsy. Such a procedure is associated with a significant risk in patients with PPFE due to pneumothorax and should be avoided. In contrast, the use of biomarkers, as discussed below, may be helpful in discriminating between IPF and PPFE, although this has not been fully explored to specifically answer the question of IPF co-existing with PPFE. The significance of finding a co-existing interstitial lung disease in patients with PPFE, as it relates to etiology and clinical outcomes, is at present not fully known. As will be discussed below, the presence of lower lobe involvement may influence the clinical course and overall prognosis. 

## 5. Pulmonary Physiology

The most consistent finding on pulmonary function testing in patients with PPFE is a decrease in the forced vital capacity (FVC) (typically below 60%) and an increase in the residual volume to total lung capacity (RV/TLC) ratio [4,12,13,14,16,25]. The RV/TLC ratio is inversely correlated with the FVC [25]. It has been proposed that a RV/TLC > 115% of normal can differentiate PPFE from other idiopathic interstitial pneumonias, which may be important in those with lower lobe involvement on imaging and concerns for a concomitant IIP [14,26]. The diffusion capacity is decreased, but to a lesser extent than is seen in idiopathic pulmonary fibrosis [4,14]. Interestingly, although chest imaging and lung histology show sub-pleural fibrosis with a loss of lung volume, patients do not experience a significant decline in exercise capacity or oxygen desaturation at rest or with exertion at the time of diagnosis [12]. At present, there are limited follow-up studies examining the course of exercise tolerance or oxygen need after diagnosis in these patients; however, follow-up studies show that with progressive disease, patients develop respiratory failure with the need for supplemental oxygen [8,13,14]. During follow-up, there is a gradual decrease in the FVC and an increase in the RV/TLC ratio in most patients [7,15,25]. For some patients, however, this change in pulmonary function testing occurs more rapidly, with a corresponding increase in their symptoms, as would be expected [27,28]. 

## 6. Challenges with Establishing a Diagnosis

The initial patients described with PPFE had a diagnosis confirmed via histologic findings, with similar findings seen on lung biopsy or at the time of autopsy [5,7,10,11]. However, patients with PPFE are at high risk for pneumothorax during their disease course and this is particularly true after undergoing lung biopsy, wherein a persistent pneumothorax may occur, leading to increased morbidity and mortality [8,12,14,23]. In addition to a pneumothorax as a complication after undergoing a lung biopsy, there is a risk for an acute exacerbation of the PPFE with the development of acute respiratory failure, possibly due to the presence of concomitant UIP [8]. Accordingly, confirmation by surgical biopsy should be avoided in these patients. This is also true for biopsies obtained by bronchoscopic means (transbronchial or cryobiopsy), in which the risk for development of pneumothorax is also high, with the potential diagnostic yield from such biopsy being lower due to the sample size [29,30]. Recently, several studies have discussed protocols to optimally confirm a diagnosis of PPFE without the need for a surgical lung biopsy. Initial studies focused on the use of radiographic imaging in patients with radiographic findings that were consistent with PPFE [7,8]. Reddy et al. outlined radiologic diagnostic criteria for definite PPFE, consistent with and inconsistent with PPFE [7]. This information was paired with histologic findings seen on lung tissue that were also graded as definite, consistent with, or inconsistent with PPFE. One limitation of this study was the requirement for lung tissue to confirm the diagnosis of PPFE. However, the development of correlations between radiographic imaging and histology, as discussed in this study, is useful in strengthening the conclusions derived from chest imaging in order to obviate the need for lung sampling to make a diagnosis of PPFE in the future. Another limitation of this study is that there was no exploration of correlation between clinical symptoms or lung physiology in relation to histologic findings or chest imaging in their patients. Subsequently, Enomoto et al. described their approach to the clinical diagnosis of PPFE in their multicenter study [8]. In this study, PPFE was diagnosed based on chest CT imaging alone using the same grading system as utilized in the study by Reddy et al. [7,8]. In contrast to the study by Reddy et al., this study also required the presence of radiologic progression to confirm the diagnosis of PPFE in order to specifically exclude patients with apical cap disease, a benign process [9]. One limitation of this study is the requirement for radiologic progression to confirm a diagnosis of PPFE. As PPFE is a slowly progressive disease, a delay in a confirmatory diagnosis awaiting radiologic progression limits the frequency of patient follow-up and serial testing, initiation of effective therapy, or inclusion in clinical trials. Finally, Watanabe et al. have proposed a diagnostic algorithm for the diagnosis of PPFE in those with or without a surgical lung biopsy (SLB) [26]. Similar to previous studies, patients were placed into diagnostic groups based on imaging and available lung histology, but unique to this study was the inclusion of clinical features in their diagnostic algorithm [26]. In those with consistent histologic findings of PPFE and upper lobe findings of subpleural consolidation and fibrosis, a definite diagnosis of PPFE was made. However, the presence of lower lobe findings required discussion by a multidisciplinary group (MDG). Unique to this study is the inclusion of the diagnostic groups of radiologically probable PPFE and radiologically and physiologically probable PPFE that requires the presence of respiratory symptoms or respiratory symptoms as well as the typical physiologic abnormalities of low BMI, decreased FVC, and an increase in the RV/TLC ratio, respectively [26]. Such an algorithm appears appropriate as it includes available radiologic, clinical, and physiologic data and does not require the need for disease progression for a confirmatory diagnosis. Such an approach at present is considered the standard of care, wherein PPFE is diagnosed based on clinical findings and consistent imaging. Similar to the protocols used in the diagnosis of other IIP’s, use of a MDG that include clinicians, radiologists, and pathologists has been proposed for the diagnosis of PPFE [26,31]. The use of such an approach was detailed in a review of a nationwide database in Japan examining clinical and radiologic features of MDG-confirmed cases of PPFE [31]. Unfortunately, at present, apart from symptoms, changes in lung physiology, and radiographic findings, with the high risk for lung biopsy for histologic analysis, there are no blood tests available to make a diagnosis of PPFE. Such a lack of a diagnostic biomarker, as will be discussed below, limits conclusively making a diagnosis of PPFE and limits the ability to identify patients with the risk of rapid progression or poor prognosis. 

## 7. Biomarkers

The availability of a biomarker to assist in the diagnosis of PPFE would be extremely helpful, both for diagnosis and to assist in identifying novel specific therapies. At present, no such biomarker exists to definitively diagnose PPFE. Several studies have examined potential biomarkers, including surfactant protein D (SPD), Krebs von den Lungen-6 (KL6), urinary desmosines, and latent transforming growth factor-β binding protein 4 (LTBP-4) [14,31,32,33,34,35,36](Table 2). These studies suggest that these biomarkers may be helpful in determining the diagnosis of PPFE or in assessing prognosis and clinical outcomes. As there is a wide variation in the rate of progression and in mortality in PPFE patients, the use of biomarkers to identify those at risk for rapid progression of the disease would be useful in targeting therapeutic trials for this specific patient group. As will be described below, most of the studies on potential biomarkers in PPFE have examined them in patients with idiopathic PPFE, however there may be extension of these findings to patients with PPFE from defined causes, including post-HSCT patients. No study to date has examined biomarkers in patients with PPFE after HSCT. Although promising, at present, larger confirmatory testing is required before any biomarker is clinically used in the diagnosis or follow-up in PPFE.

A small study examined urinary desmosine levels in patients with biopsy-proven PPFE compared to levels in controls and in patients with idiopathic pulmonary fibrosis (IPF) or chronic obstructive disease (COPD) [33]. Desmosines are breakdown products of elastase, and it was hypothesized that urinary levels of desmosines would be higher in PPFE patients where active fibrosis is occurring, as evidenced by the extensive elastic fibers seen on histology [33]. In addition, rates of fibrosis are higher in patients with PPFE compared to those with IPF, thereby suggesting a potential tool to differentiate between the two diagnoses [33]. Results from this study confirmed the researchers’ hypothesis as urinary desmosines levels were significantly higher in PPFE patients compared to controls and those with COPD. In addition, desmosines levels were higher in PPFE patients compared to those with IPF, with urinary desmosine levels able to discriminate between those with PPFE and those with IPF. In contrast, levels of urinary desmosine levels did not correlate with clinical parameters (e.g., BMI), physiologic variables (pulmonary function testing, PaO2, or 6 min walk distance), or severity of disease [33]. One limitation of the study was that only a single time point was examined, with the question remaining of whether urinary desmosines can be utilized longitudinally to assess the risk of disease progression or the long-term prognosis. At present, measurements of urinary desmosines may be clinically useful to differentiate PPFE from IPF in patients with lower lobe involvement or imaging findings that are suggestive of IPF, but further studies are necessary. 

Levels of KL-6 have been shown to be useful in the diagnosis of PPFE and in differentiating PPFE from other forms of idiopathic interstitial pneumonias (IIP), as well as in prognostics. The study by Oyama et al. examined levels of KL-6 between those with PPFE and IPF and found that levels of KL-6 were over 3-fold higher in IPF patients compared to those with PPFE [33]. However, there was no correlation between KL-6 levels and clinical characteristics or physiologic variables in the PPFE patients that were examined in that study. This specific question was examined in a study by Ishii et al., who showed that elevated levels of KL-6 in PPFE patients were associated with a median survival that was 24 months shorter than in those with KL-6 levels under 600 U/mL during a follow-up period of over 5 years [31]. Similar findings were reported by Kinoshita et al., wherein KL-6 levels over 550 U/mL were associated with an overall poorer prognosis [34]. Finally, d’Alessandro et al. examined the prognostic ability of serum KL-6 levels in systemic sclerosis patients with PPFE. Those with increasing serum KL-6 levels also had significant declines in their FVC and FEV-1 levels, thereby suggesting a clinical progression of PPFE in that group of patients [32]. One strength of this study is that KL-6 levels were measured serially over a six-year timeframe, thereby increasing the clinical relevance of their results. In contrast to the above studies, Sato and colleagues did not find a difference in the levels of KL-6 in PPFE patients in a small study [35]. 

Recently, Kinoshita et al. examined the use of LTBP-4 as a biomarker in patients with idiopathic PPFE [34]. Levels of LTBP-4 are thought to correlate with ongoing elastogenesis [34]. In this study, levels of LTBP-4 were determined in both lung tissue and in serum from both clinically diagnosed PPFE patients or those with confirmed PPFE via lung biopsy. Comparisons were made to levels of LTBP-4 in those with IPF or in healthy control subjects. Serum levels of LTBP-4 were significantly higher in PPFE patients compared to controls and were 50% higher in PPFE patients compared to those with IPF, although this did not reach statistical significance, likely due to the overall small number of patients in this study [34]. In addition, lung levels of LTBP-4 were over 2-fold higher in patients with PPFE compared to those with IPF [34]. Interestingly, IPF patients with the presence of co-existing upper lobe PPFE had higher levels of LTBP-4 in the serum compared to IPF patients without co-existing PPFE, again suggesting that LTBP-4 may be a more specific biomarker for PPFE. Regarding correlation with clinical parameters and survival, higher serum LTBP-4 levels trended with the findings of a more rapid change in the FVC and the residual volume to total lung capacity (RV/TLC) ratio, and patients with higher LTBP-4 levels had over a 20% worse survival rate over time [34]. When comparing the discriminative value of LTBP-4 and KL-6 in differentiating between PPFE and IPF, and both of those conditions versus control patients, KL-6 was found to be more useful for the diagnosis of IPF and LTBP-4 more useful for PPFE patients [34]. A limitation of this study is that only a one-time measurement of serum LTBP-4 was assessed. Similar to KL-6, it would be interesting to determine if serial LTBF-4 measurements over time can predict the rate of progression of PPFE or overall longevity and survival. 

Finally, serum levels of surfactant protein D (SPD) have been shown to be elevated in PPFE patients in several small studies [8,35,36,37,38]. Although elevated, no studies have shown a correlation between SPD levels and baseline physiologic values or demographics in PPFE. In a cluster analysis designed to identify prognostic factors for PPFE, there was a trend toward higher SPD levels in the group of patients with the highest mortality rates [37]. Further studies are necessary in larger cohorts of patients to further define the role of SPD testing in PPFE and to determine if serial testing can identify patients at risk for disease progression or a decreased length of survival.

The above studies examining potential biomarkers in PPFE were mostly performed in patients with the idiopathic form of the disease. However, as the pathologic findings seen on lung biopsy and autopsy samples do not differ between those with identified etiologies for PPFE, such as post-HSCT patients, and those with the idiopathic form, overall, this suggests that similar biomarkers may be useful in the diagnosis and management of PPFE in both of these patient groups.

## 8. Treatment, Clinical Course, and Prognosis

To date, no pharmacologic treatment for PPFE has been identified to be helpful. Several case series describe patients that were treated with corticosteroids, but overall, corticosteroids were not felt to be beneficial in slowing down the disease progress or changing outcomes in those studies [4,12,13,14,18]. Similarly, the use of other immunosuppressants and adjunctive treatments, including cyclophosphamide, N-acetylcysteine, and azithromycin, has been attempted, but no reports have indicated a consistent response to these treatments [12,13,14]. Recently, the use of nintedanib was described in a small series of patients with either idiopathic or secondary PPFE [13,39]. In the study by Nasser et al., patients that received nintedanib had a significant decrease in the decline of the FVC over time, although upper lobe volume loss, as measured by CT, did not differ between those treated with nintedanib or those off medications [39]. Future studies are necessary to identify and evaluate novel pharmacologic treatments for PPFE. For those with progressive disease and respiratory failure, the only treatment option that has shown to be effective is lung transplantation [12,13,14]

A diagnosis of PPFE carries an overall poor prognosis, with 5-year survival rates of only 23.3–58.9% [7,8,14,24]. Two clinical scenarios have been described in the course of PPFE, that of a slow progression of disease and one with a more precipitous decline, with the former being the typical clinical course [4,30]. Progression is typically manifested in ongoing upper lobe volume loss with flattening of the thoracic cage and decreasing BMI, with an associated increase in shortness of breath and a decline in the FVC. The presence of lower lobe involvement and/or the presence of a co-existing interstitial lung disease is associated with a more rapid disease progression and shorter survival duration [39]. Recently, studies have been published of scoring systems to assist in assessing the prognosis of individNotuals with PPFE. The gender–age–physiology (GAP) model is widely used as a prognostic tool in idiopathic pulmonary fibrosis and was evaluated by Shioya et al. in patients with PPFE [40]. A higher GAP model predicted a worse prognosis in that study [40]. Kinoshita et al. recently published a PPFE prognostic model using variables shown to be significant in PPFE patients [9]. Using two large cohort samples of patients with PPFE, Kinoshita and colleagues identified four variables in a multivariate Cox regression model that were predictive of mortality in PPFE: percent of forced vital capacity, history of pneumothorax, the presence of interstitial lung disease in the lower lobes, and serum KL-6 levels [9]. Using these four variables, a point system was derived, and patients were placed into one of three groups based on a total score. This scoring system could separate patients into unique groups of estimated survival. Such scoring system will be extremely useful in the future to better inform patients and families of the expected clinical course and for enrollment in future clinical treatment trials.

## 9. Conclusions

Pleuroparenchymal fibroelastosis (PPFE) is more commonly considered an idiopathic disease, but it has been rarely observed in patients who have previously undergone hematopoietic stem cell transplant (HSCT). In both the idiopathic forms and after HSCT, PPFE remains poorly understood. After HSCT, PPFE is a very late complication, occurring on average over six years from the time of transplant. Based on the present studies, it seems to be a form of graft versus host disease, with a strong co-association with bronchiolitis obliterans syndrome, the primary outcome of pulmonary-associated graft versus host disease; however, a direct pathophysiologic link has not been made. Advances in diagnostic algorithms to diagnose PPFE without the need for histologic analysis of lung tissue have been made that will reduce patient morbidity and mortality by avoiding a post-lung biopsy pneumothorax. Unfortunately, no effective therapy has been found for PPFE, either in idiopathic or post-HSCT cases, although the disease progression is slow for most patients.

## Figures and Tables

**Table 1 biomedicines-11-00924-t001:** Literature of PPFE after HSCT.

Von der Thusen J, et al., 2011 [10]	Describes initial case reports of PPFE after HSCT.
Takeuchi Y, et al., 2015 [11]	Discusses pathologic findings of PPFE in post-HSCT patients as well as other interstitial lung diseases found in these patients.
Bondeelle L, et al., 2020 [13]	Excellent overview of the clinical characteristics, potential etiologies, imaging, pathology, and clinical course of PPFE after HSCT.
Namkoong H, et al., 2017 [16]	Discusses the etiology of the uncommon complication of restrictive lung volumes after HSCT.
Meignin V, et al., 2018 [17]	Focuses on the histopathologic findings of pulmonary complications after HSCT with clinical characteristics and patient demographics.
Higo, H, et al., 2019 [18]	Comprehensive review of the literature to determine the etiology for PPFE after HSCT.
Busmail, A 2022 [19]	An excellent overview of the pulmonary complications after HSCT.

**Table 2 biomedicines-11-00924-t002:** Biomarkers for PPFE.

Biomarker	Relative level	Notes
Desmosine	PPFE > IPF > Normal	No correlation of level to clinical markers. May assist in differentiating co-existing IPF with PPFE in lower lobes.
Krebs von den Lungen-6 (KL-6)	IPF > PPFE	May assist in predicting survival.
Latent transforming growth factor-β binding protein 4 (LTBP-4)	PPFE > IPF > Normal	Correlative to decline in FVC. May assist in predicting survival.
Surfactant Protein D	PPFE > Normal	Higher levels associated with higher mortality rates.

## Data Availability

No new data created for this study.

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
