# Peer review of "Pleuroparenchymal Fibroelastosis: A Review with a Focus on a Non-Infectious Complications after Hematopoietic Stem Cell Transplant"

_biomedicines, 2023, doi:10.3390/biomedicines11030924_

Round 1
Reviewer 1 Report
Dear authors
The manuscript is well-written and interesting in its area of research. I propose to add a table and a graph describing the findings of this study.
kind regards
Author Response
I would like to thank the Reviewer for their time in reviewing the manuscript and for their helpful suggestions in improving the manuscript. I am delighted that you found the manuscript to be interesting. Per your suggestion I have included two tables in the revised manuscript; one outlining manuscripts published regarding PPFE after HSCT and another summarizing results of biomarker studies in PPFE. I hope that the Reviewer finds the revised manuscript acceptable for publication.
Reviewer 2 Report
This is a very interesting review on pleuropulmonary elastosis
2 minor comments
1 how to differentiate it from the IPF?
2 a table with the studies reviewed so that the reader had a quicker image on the rarity of the disease and its closer link to stem cell transplantation
Author Response
I would like to thank the Reviewer for their time in reviewing the manuscript and for their helpful suggestions in improving the manuscript. I am delighted that you found the manuscript to be interesting. Per your suggestion I have included additional description on how to differentiate IPF and PPFE. This is a very interesting point and continues to be a challenge at this point. In addition, I have now included a table outlining manuscripts published regarding PPFE after HSCT to assist the reader in reviewing this topic. I hope that the Reviewer finds the revised manuscript acceptable for publication.
Reviewer 3 Report
This review discussed what is currently known regarding PPFE as a complication after HSCT highlighting discussions regarding establishing a diagnosis, potential biomarkers, current treatments, and clinical outcomes and prognosis.I think the author gave a detailed introduction to this field.
Author Response
I would like to thank the Reviewer for their time in reviewing the manuscript and for their helpful suggestions in improving the manuscript. I am delighted that you found the manuscript to be a detailed introduction. In the revised manuscript, in response to helpful comments from the Reviewers, I have included two tables; one outlining the manuscripts published regarding PPFE after HSCT and another summarizing results of biomarker studies in PPFE. I have also expanded the discussions regarding the differentiating IPF from PPFE. I hope that the Reviewer finds the revised manuscript acceptable for publication.